



# Accretionary prism deformation and fluid migration caused by slow earthquakes in the Nankai subduction zone

Takashi Tonegawa[1], Takeshi Akuhara[2], Yusuke Yamashita[3,] Hiroko Sugioka[4], Masanao Shinohara[2], Shunsuke Takemura[2], Takeshi Tsuji[5]

[1]Japan Agency for Marine-Earth Science and Technology (JAMSTEC), Yokohama, 236-0001 Japan
[2]Earthquake Research Institute, The University of Tokyo, Tokyo, 113-0032, Japan
[3]Disaster Prevention Research Institute, Kyoto University, Miyazaki, 889-2161, Japan
[4]Department of Planetology, Graduate School of Science, Kobe University, Kobe, 657-8501, Japan
[5]Department of Systems Innovation, Faculty of Engineering, The University of Tokyo, Tokyo, 113-8656, Japan

*Correspondence to*: Takashi Tonegawa (tonegawa@jamstec.go.jp)

**Abstract.** Slow earthquakes induce structural deformation around their source regions and can also be linked to fluid migrations. Both of these phenomena potentially induce temporal variations in the seismic structure; how these two factors behave for a slow earthquake episode remains unknown. In this study, we applied the ambient noise correlation technique to the continuous records acquired at the seafloor in the Nankai subduction zone, and investigate the changes in seismic velocity (*dv/v*) and heterogeneous (correlation coefficient (CC) representing heterogeneity changes) structures before and after the slow earthquake activity that occurred around the shallow plate interface from the end of 2020. As a result, temporal variations in *dv/v* and CC show different patterns for the slow earthquake episode. The *dv/v* variations show a step-like reduction and the reduced velocity was not recovered to the original level until the end of the observation period, whereas the CC variations show transient reductions and were recovered to the original level after the episode. Thus, the *dv/v* reflects the variation in the aspect ratio of pre-existing cracks and/or newly created cracks due to sediment deformation, where the extensional stresses normal to the trough were induced by the slips of the slow earthquakes, and the updated crack condition persisted even after the episode. We suggest that the CC variations correspond to transient fluid migrations from the source regions to shallow depths, activated by the fracturing of fluid caprocks that resulted from slow earthquakes. Our study indicates that monitoring these two quantities provides useful information to understand the variations in the subsurface structure due to slow earthquakes.

**Short summary.** This study demonstrates that for slow earthquakes in the shallow Nankai subduction zone, (1) the velocity reductions in the accretionary prism reflect the stress field changes due to the slow earthquakes, and (2) the temporal variations in the heterogeneous structure correspond to upward fluid migrations from the source region of the slow earthquakes.





## 1 Introduction

Slow earthquakes occur at shallow plate interfaces in the Nankai subduction zone in southwestern Japan, and their characteristics have been investigated from interdisciplinary viewpoints, such as seismic and geodetic observations, geology, experiments, and seismic structures (e.g., Takemura et al. 2023). The occurrences of tremors, very low frequency earthquakes

(VLFEs), and slow slip events (SSEs) have been detected by multivariant geophysical records from seismometers in both land and sea areas (e.g., Obara and Ito, 2005; Ito and Obara, 2006; Asano et al., 2008; Sugioka et al., 2012; Kaneko et al., 2017; Takemura et al., 2019; Yabe et al., 2019; Baba et al., 2020; Ogiso et al., 2022; Tamaribuchi et al., 2022; Yamamoto et al., 2022), pore pressure measurements in the boreholes at the seafloor (Araki et al., 2017; Nakano et al., 2017; Ariyoshi et al., 2021), and seafloor geodetic observations using Global Navigation Satellite System (GNSS)–acoustic ranging combined with

a seafloor positioning system (GNSS-A) (Yokota and Ishikawa, 2020). Tremors and VLFEs are sporadically distributed along the strike direction of the Nankai Trough, and they occur at the plate interface at depths less than ~10 km (Obara and Kato, 2016). Although the recurrence intervals of these phenomena vary regionally along the Nankai Trough, they are approximately 1–5 years (e.g., Baba et al., 2020).

Fluids are related to the generations of slow earthquakes. This linkage has been documented for slow earthquakes occurring at

the both deep and shallow plate interfaces in various subduction zones. Seismological techniques, including tomography and receiver functions, have revealed that the subducting oceanic crust near the region where deep tremors occur shows specific seismic velocities, such as low P wave velocity (Vp) in the Nankai subduction zone (Shelly et al., 2007) and high Vp/Vs in the Cascadia subduction zone (Gosselin et al., 2020). A tomographic study suggested that low-frequency earthquakes occur under undrained conditions within the subducting oceanic crust (Nakajima and Hasegawa, 2016). Details on other subduction

zones have been reviewed elsewhere (Audet and Kim, 2016). Pore pressure waves along the plate interface may influence the migration of tremor excitations (Cruz-Atienza et al., 2018). Such fluid migrations to different sites are also related to slow earthquake generations (e.g., Nakajima and Uchida, 2018; Ito and Nakajima, 2024). These seismological features indicate the prevalence of fluids with high pore pressure within the oceanic crust, which mechanically reduces the shear strength of the fault.

In the cases of shallow slow earthquakes, evidence of fluids and their migrations has been reported. Seismic exploration surveys have reported the presence of a low velocity layer within the accretionary prism in the central Nankai subduction zone (Park et al., 2010; Kamei et al., 2012), and this is linked to the generation of VLFEs through the presence of fluids (Kitajima and Saffer, 2012; Tsuji et al., 2014; Tonegawa et al., 2017; Akuhara et al., 2020). Another exploration study in the Hikurangi subduction zone indicated that sediment lenses with low velocity left behind subducting seamounts are linked to SSEs (Bangs

et al. 2023). In the Hyuga-nada region of the western Nankai subduction zone, low seismic velocity layers have been detected within marine sediments above the regions where tremors occur (Akuhara et al., 2023a). Arai et al. (2023) has successfully imaged the vertical low velocity zones connecting between the plate boundary and seafloor, and found a spatial correlation between the vertical conduits and the distribution of slow earthquakes.



In order to unveil the relationship between fluids and slow earthquake generations, the temporal changes in the seismological
structure around the region of slow earthquakes should be investigated. Using the continuous records of permanently-deployed
ocean bottom seismometers (OBSs) (Dense Oceanfloor Network system for Earthquakes and Tsunamis, DONET) (Fig. 1)
(Kaneda et al., 2015; Kawaguchi et al., 2015; Aoi et al., 2020) for the observation periods >10 years, which was acquired in
the Nankai subduction zone, Tonegawa et al. (2022) found that the heterogeneous structure shows temporal changes associated
with slow earthquakes, and interpreted that transient fluid migrations alter the fractional fluctuation of wave velocity in the
heterogeneous structure within the sediments. In particular, the slow earthquake activity that started at the end of 2020 showed
the temporal changes in seismic velocity and heterogeneous structure. However, these variations after the slow earthquake
activity have not been investigated in details because post-event data were not sufficiently available. Moreover, the mapping
of temporal variations in the seismic structure was limited to the DONET1 area (Tonegawa et al., 2022).

In this study, using the continuous records of the DONET and temporary OBSs, we estimate the temporal changes in
seismological structure with the ambient noise correlation technique, and further investigate the linkage between fluids and
slow earthquakes that started from the end 2020. Although the recurrence intervals of SSEs and both tremors and VLFEs in
this area are approximately 1 year (Araki et al., 2017) and 4–5 years (e.g., Takemura et al., 2019; Baba et al., 2020), respectively,
these three phenomena occurred during this episode. The tremors started from December 5, 2020, and their activities became
high around December 11 and 28  (Tamaribuchi et al., 2022; Ogiso and Tamaribuchi, 2022) (Fig. 1). Relatively weak activities
continued until the middle of January 2021, and a few tremors occurred until the beginning of February. Tremors also occurred
in the gap area between DONET1 and DONET2, and the distribution of temporary OBSs covers the spatial gap between the
DONET stations (Fig. 1). Their observation periods contain the timing of the episode and its recovery time sufficiently. Thus,
the integration of the temporary OBS data with the DONET data allows us to investigate the spatio-temporal variations in the
seismic structure associated with the slow earthquake activity in detail.





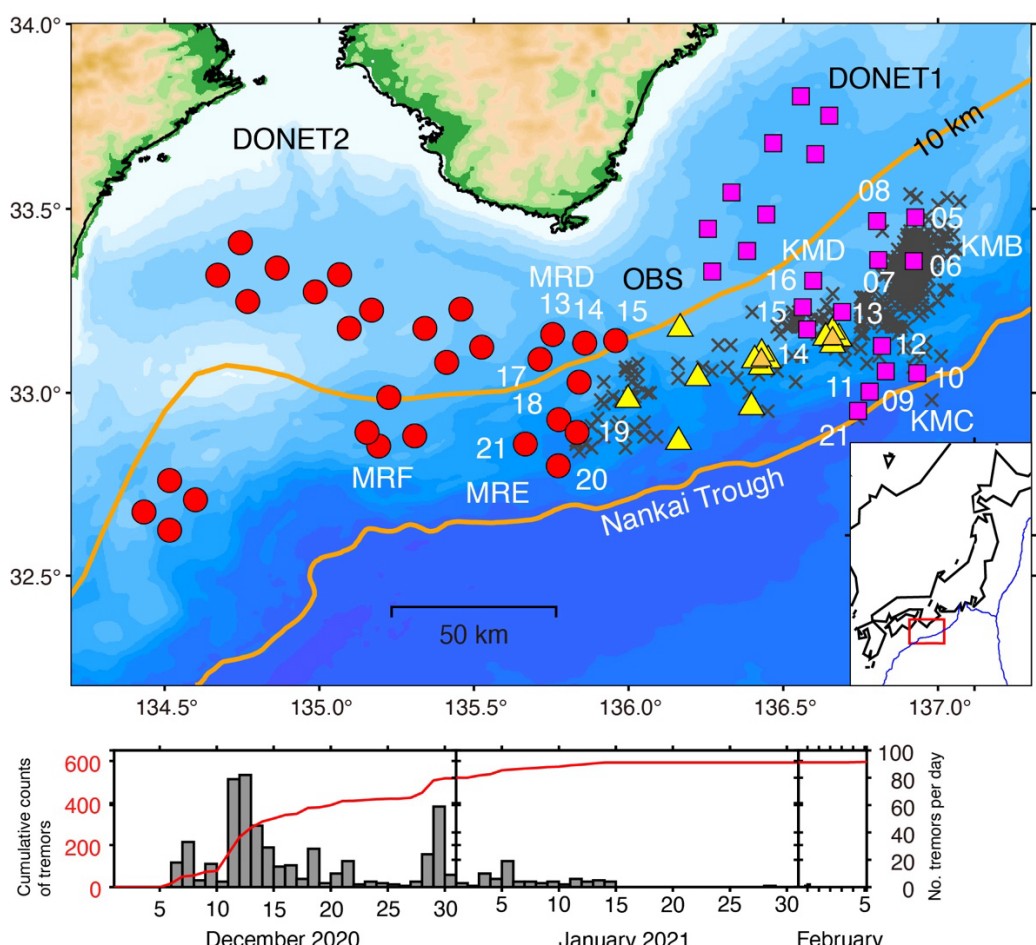

**Figure 1: (Top) Map showing the locations of stations. Circles and triangles indicate the stations of DONET1 and DONET2, respectively, and squares represent temporary OBSs. The two lines correspond to the contour lines of 0 km (Nankai Trough) and 10 km for the Philippine Sea Plate (Baba et al. 2002; Hirose et al. 2008). Crosses indicate the locations of tremors that occurred**
**between December 1st 2020 and February 5th, 2021 (Tamaribuchi et al. 2022), for which tremors with spatial errors ≤0.1° are plotted and 16% of tremors are preserved from the original catalog. (Bottom) Frequency of tremors that occurred between December 1, 2020, and February 5, 2021, with the cumulative numbers by the solid line and counts per day by the histogram. The tremors are the same as those used in the top panel.**

**2 Data**

We used the continuous records in the vertical components of the 49 stations of DONET (Fig. 1). However, those from the southern part were primarily considered because temporal changes due to the slow earthquakes were limited to this region (Tonegawa et al., 2022). DONET contains the eastern (DONET1) and western (DONET2) cabled networks, which are deployed at water depths of 1,000–4,400 m with a station spacing of 10–30 km and are installed at the end of 2010 and 2014,



respectively. All stations include a broadband seismometer (Guralp CMG-3T, flat velocity response up to 360 s), which is
buried 1 m below the seafloor (Nakano et al., 2013). The locations of the DONET1 and DONET2 differ from each other by
~80 km. To fill the gap, 15 OBSs with short-period sensors (1Hz) were temporarily deployed in this area for the period between
September 2019 and May 2021, in which 10 OBSs were used for every two subarrays with a station spacing of ~2 km. In this
study, we used 2 OBSs located at the center of the two subarrays, and we used 7 OBSs in total.

## 3 Methods

We followed the method of Tonegawa et al. (2022) to calculate the cross-correlation functions (CCFs) and map the spatio-
temporal variations in seismic velocity and heterogeneous structures. However, we used temporary OBSs in this study, and
their clocks during the observation period had uncertainties. The clocks of the recorders are adjusted to the clocks of the GNSS
at the timings before and after their observations, and are linearly interpolated during the observations. In Section 3.2, we
describe the estimation of the uncertainties in the clocks during the observation periods.

### 3.1 Calculation of cross correlation functions (CCFs)

We calculated CCFs using ambient noise records (Campillo and Paul, 2003; Shapiro et al., 2005; Brenguier et al., 2007).
Ambient noise records contain energetic signals, including those of earthquakes, which are suppressed by lognormal-shaped
functions (Tonegawa et al., 2020; 2022). We used a bandpass filter of 0.5–2.0 Hz with spectral whitening. In this frequency
band, acoustic-coupled Rayleigh (ACR) waves are dominant in the ambient noise records, which propagate in the ocean and
the entire accretionary prism with a propagation velocity of 1.3–1.5 km/s in this region (Tonegawa et al., 2015). The time
window for calculating the CCFs is 600 s, and the 30-day moving average of the CCFs was calculated for everyday by stacking
them for each station pair. However, if the time length in which the lognormal-shaped functions are applied is less than 70 %
of 30 days, we discarded the 30-day CCF. The reference CCF for each station pair was also calculated by stacking the CCFs
over the entire observation period. The ACR waves directly propagating between two stations emerge in the early lag times of
the CCFs, and their scattered waves emerge in their coda parts (Fig. 2).




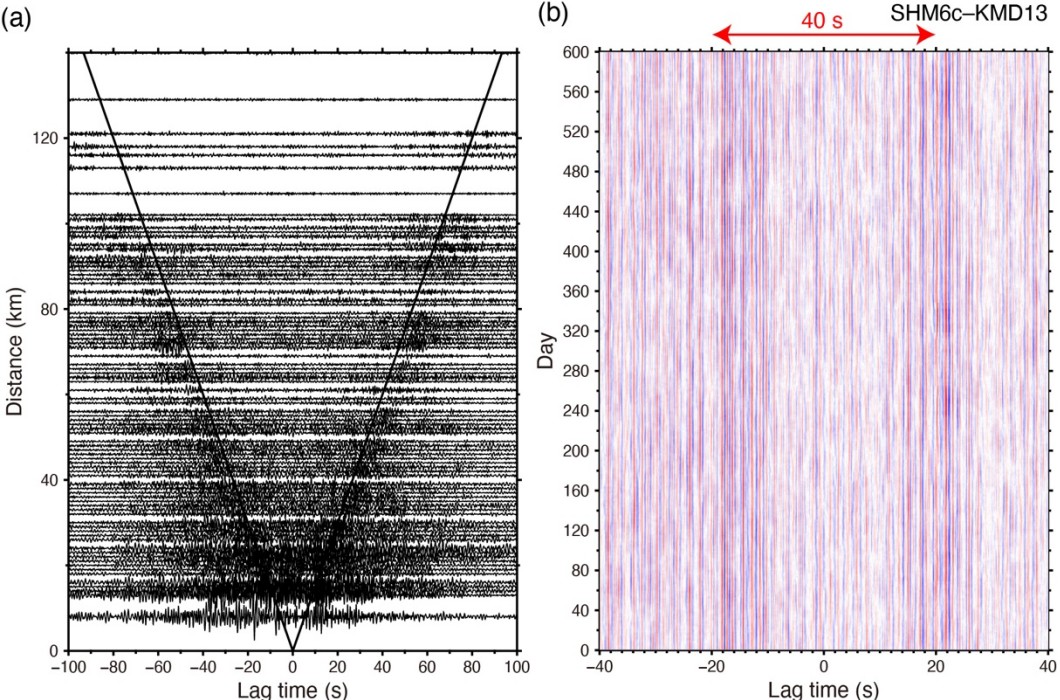

**Figure 2: Stacked CCFs at 0.5–2.0 Hz. (a) 365-day CCFs for station pairs including temporary OBSs. The reference date is April 1st,**
**2020, and the CCFs within −182–183 days from the reference date were stacked. The two lines are the reference lines for a**
**propagation velocity of 1.5 km/s. (b) CCFs at the station pair of SHM6–KMD13 for the observation period.**

## 3.2 Clock error estimations

The clocks of the recorders are linearly interpolated between the starting and ending times of the observations. However, the
deviation rate of recorder clocks from the accurate times may not be uniform through the observation period. These deviations
can be measured using CCFs with ambient noise correlations (Takeo et al., 2014). Using the CCFs calculated in the previous
section, we measure the clock deviations of the OBSs from the clocks of the DONET stations. The clocks of the DONET data
are calibrated by the GNSS signal through the submarine cables to land (e.g., Tsuji et al., 2023). We used CCFs for station
pairs between OBSs and DONET stations, and the DONET stations are selected within a distance of 50 km from each OBS.
When the CCFs are aligned as a function of the observation date, several coherent phases, including ACR waves and their
scattered waves, are observed in the positive and negative lag times over the observation periods (Fig. 2b). However, if the
clocks are deviated from the accurate times, the arrival times of the causal and acausal waves are simultaneously shifted in the
same direction towards the positive or negative lag time during the observation periods. Here, in cases where temporal
variations in the seismic velocity at the subsurface structure affect the arrival times of causal and acausal waves, the arrival
times of causal waves in the positive lag time and acausal waves in the negative lag time are shifted symmetrically with respect
to the lag time of 0 s, which is a different feature from the clock deviations to the CCFs.




We set a 2-s time window with an increment of 1 s between −20–20 s in the CCFs (Fig. 2b), and prepare the reference CCF for each station pair by stacking the CCFs between October 2019 and September 2020. In each 2-s time window, we calculate the cross-correlation functions (referred to as CF, to distinguish this function from the CCFs using ambient noise records)

between the reference CCF and individual CCFs. In a given time window, if the cross correlation coefficients of the CFs are greater than 0.9 for more than 85% of the observation period, the time window is assumed to contains coherent signals over the observation period, and the delay times of the CFs are used as the clock deviation. The obtained temporal variation of the clock deviation from the starting time of the observation period may be slightly shifted from 0 s in lag time, because the coherent signals between the reference CCF and individual CCFs may have phase differences. Therefore, the time difference

of a coherent signal averaged over the first 10 days is subtracted from the time differences of the subsequent days. This processing is repeated for different pairs between each OBS and the available DONET stations within a distance of 50 km from the OBS. On each day, the median value of the obtained delay times is defined as the clock deviation, and the first and third quartiles are considered as the uncertainties (Figs. 3 and S1).

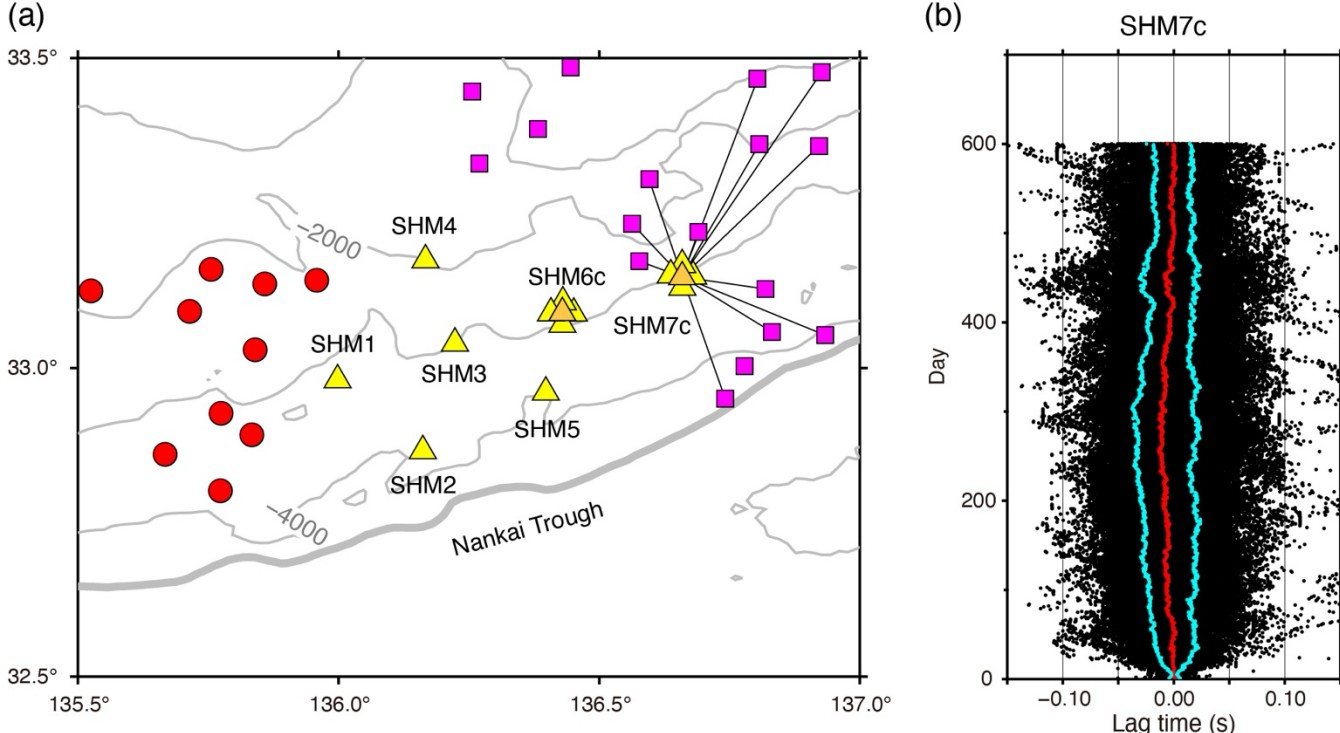

**Figure 3: Clock deviations at SHM7c. (a) The locations of the pairs between SHM7c and DONET stations used for measuring the clock deviations, which are connected by solid lines. Other symbols are the same as those used in Fig. 1. (b) The estimated clock deviations at SHM7c. Dots represent the measured clock deviations for SHM7c. The left-side and right-side lines indicate the first and third quartiles, respectively, and the center line represents the median values.**



### 3.3 Estimations of *dv/v* and CC

For each 30-day CCF, we applied the stretching technique for estimating the changes in seismic velocity (*dv/v*) and cross-correlation coefficients (CC) (Sens-Schönfelder and Wegler, 2006; Obermann et al., 2013). The searching range of the stretch (*dt/t*) is −0.4–0.4 % with an increment of 0.01 %, and *dv/v = –dt/t*. Using the causal and acausal parts (positive and negative lag times) of the CCFs, we also estimated the CC with the 50-s segments of the reference CCF and 30-day CCFs that were stretched by the obtained *dv/v*. The starting times of the segment are the arrival times of the ACR waves in the positive and negative lag times, which was estimated by dividing the separation distance of two stations by a propagation velocity of 1.5 km/s. After correcting the individual CCFs using the obtained *dv/v*, we calculated the CCs between the reference and the corrected CCFs.

The uncertainties in *dv/v* and CC due to the clock deviations are estimated by a bootstrapping technique. The *dv/v* and CC variations were estimated using the 30-day CCFs every day (−14th –+15th days from the reference days), and the clock deviations were also estimated for the OBSs every day. We randomly selected a value between the first and third quartiles, and time-shifted the CCFs using this value as the clock deviation. Using the time-shifted CCFs, we estimated *dv/v* and CC, and repeated this process 100 times to obtain *dv/v* and CC uncertainties. The results for the node KMC and OBSs are summarized in Figs. 4 and S2. *dv/v* and CC could not be estimated at several pairs of stations in node MRE (Fig. S2d). This is because we discarded the 30-day CCFs for the reasons mentioned in Section 3.1, and we did not use the CCFs for *dv/v* and CC mappings, which will be explained in the subsequent section.

Because we use ACR waves, the depth sensitivities of the *dv/v* and CC obtained in this study are approximately 1 km from the seafloor (Tonegawa et al., 2022). Moreover, since the stretching technique measures the *dt/t* for scattered waves in the coda part of the CCFs, the obtained *dv/v* reflects relatively wide horizontal areas around two stations. In contrast, because the CC potentially varies with amplitude variations within a short time segment of the CCFs, the obtained CC reflects the temporal variations of the heterogeneous structure in relatively horizontally localized areas where the scattered waves within the short time segment sample. Indeed, an experimental study suggested that the CC variation was more sensitive than *dv/v* variations to fluid injection (Théry et al., 2020).





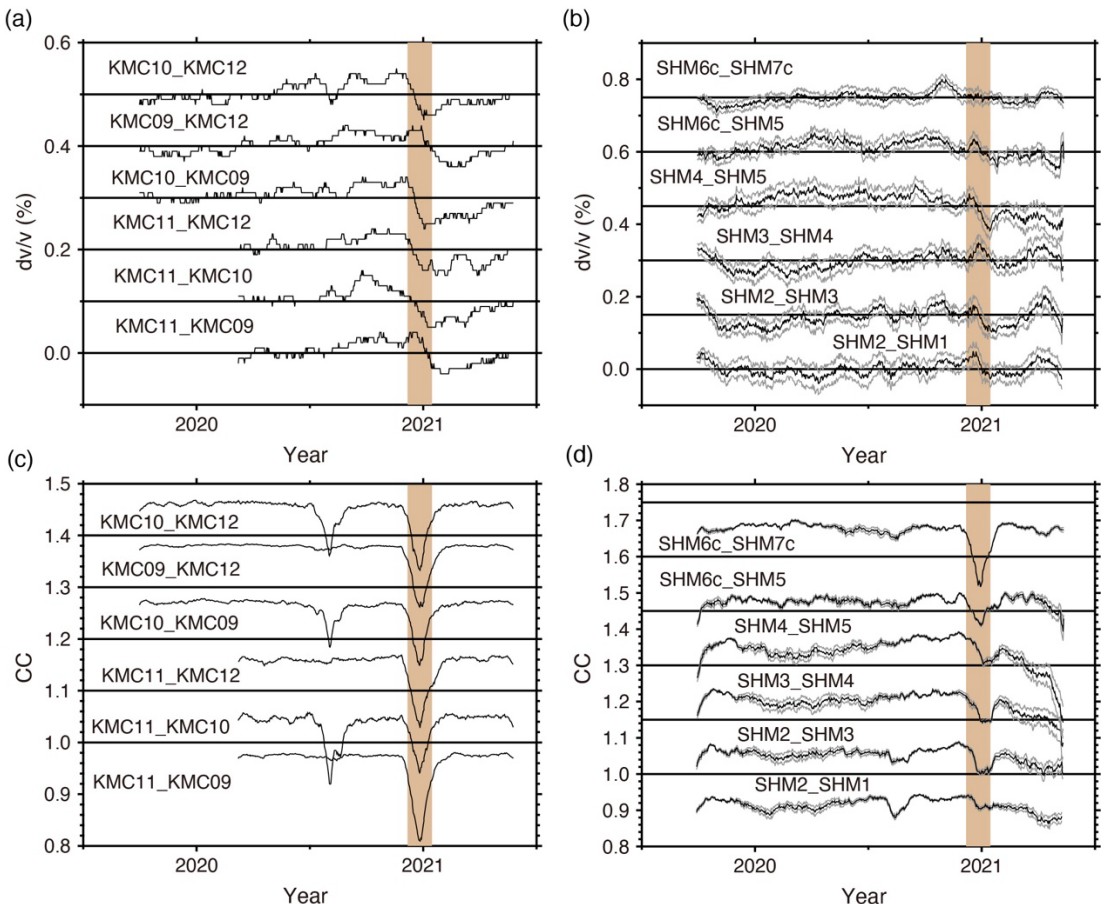

**Figure 4: (a)** *dv/v* **variations for node KMC. The shaded region represents the timing of the slow earthquake activity. (b) Same as panel (a), but for results for the OBSs. Upper and lower lines indicate the uncertainties due to clock deviations. (c) Same as panel (a), but for CC variations. (d) Same as panel (b), but for CC variations.**

## 3.4 Spatial mapping of velocity change and seismic scattering coefficient

To estimate the region where the velocity and heterogeneous structure change, we map velocity changes and scattering coefficient changes (SCCs: *Δg*) using a method described previously (Obermann et al., 2013; 2014; 2015; Sánchez-Pastor et al., 2018; Hirose et al., 2019; 2020; Tonegawa et al., 2022). We limited the mapping of *dv/v* and *Δg* with available stations in nodes KMB, KMC, KMD, MRD, MRE, and temporary OBSs (Fig. 1), because the *dv/v* and CC variations are limited to the southern part of the used stations. The tremor activities started on December 6, 2020, and finished on January 14, 2021. We used the reference values for the *dv/v* and CC averaged over November 21–30, 2020, and selected the values of *dv/v* and CC for Periods (1)–(6); (1) November 25, 2020, (2) December 10, 2020, (3) December 20, 2020, (4) December 30, 2020, (5) January 10, 2021, and (6) January 20, 2021. Because the used CCFs are stacked over 30 days, obtained results on *dv/v* and CC





contain the effects between the $-14^{th} - +15^{th}$ days from the reference dates. We prepare a dataset of $F_i$ defined as the difference of $dv/v$ or CC between the values of Periods (1)–(6) and the reference values. The observed $F_i$ can be expressed as

$$\boldsymbol{F} = \boldsymbol{Gm} \qquad (1),$$

where $\mathbf{F}$ is the vector of $F_i$ ($i$=1…n) and $\mathbf{m}$ consists of parameters (velocity changes or $\Delta g$) that we estimate for each cell $j$. $\mathbf{G}$ is a matrix with each component,

$$G_{ij} = \frac{\Delta s}{t} K_{ij} \qquad (2),$$

and

$$G_{ij} = \frac{c\Delta s}{2} K_{ij} \qquad (3),$$

respectively, where $\Delta s$ is the surface area of the grids, $c$ is the apparent velocity of the wave, and $t$ is the center of the time interval in the coda part. $K_{ij}$ is the sensitivity kernel for station pair $i$ in cell $j$, and $K$ in $\mathbf{x_0}$ can be described by the positions of two stations, $\mathbf{s_1}$ and $\mathbf{s_2}$, as

$$K(\boldsymbol{s_1}, \boldsymbol{s_2}, \boldsymbol{x_0}, t) = \frac{\int_0^t p(\boldsymbol{s_1}, x_0, u) p(x_0, \boldsymbol{s_2}, t-u) du}{p(\boldsymbol{s_1}, \boldsymbol{s_2}, t)} \qquad (4).$$

$p(\mathbf{s_1}, \mathbf{s_2}, t)$ is the probability that the waves have propagated from $\mathbf{s_1}$ to $\mathbf{s_2}$ at time $t$, which can be approximated by the intensity of the wavefield from $\mathbf{s_1}$ to $\mathbf{s_2}$ at time $t$. The analytic two-dimensional solution of the radiative transfer for isotropic scattering in the intensity propagator is given by

220 $$p(r,t) = \frac{e^{-\frac{ct}{l}}}{2\pi r} \delta(ct-r) + \frac{1}{2\pi lct} \left(1 - \frac{r^2}{c^2t^2}\right)^{-\frac{1}{2}} exp^{\left[\sqrt{c^2t^2-r^2}-ct\right]/l} \Theta(ct-r) \qquad (5),$$

where $r$ is the distance between source and receiver, $l$ is the transport mean free path, and $\Theta(x)$ is the Heaviside step function. The theoretical $F^{th}_i$ for velocity changes and $\Delta g$ in $\mathbf{x_0}$ can be computed by

$$F_i^{th}(\boldsymbol{s_1}, \boldsymbol{s_2}, \boldsymbol{x_0}, t) = \frac{\Delta g}{t} K(\boldsymbol{s_1}, \boldsymbol{s_2}, \boldsymbol{x_0}, t), \qquad (6)$$

and

$$F_i^{th}(\boldsymbol{s_1}, \boldsymbol{s_2}, \boldsymbol{x_0}, t) = \frac{c\Delta g}{2} K(\boldsymbol{s_1}, \boldsymbol{s_2}, \boldsymbol{x_0}, t) \qquad (7),$$

respectively. In this study, we use $c$ = 1.5 km/s for the ACR wave, and a transport mean free path of 10.8 km (Tonegawa et al., 2022). We used a non-linear inversion technique, the simulated annealing (Sen and Stoffa, 1995), with setting $\Delta s$ as 25 km$^2$ (=



5 km × 5 km) to find the best solutions of *dv/v* and *Δg* at each grid  (Tonegawa et al., 2022), in which the difference between

$F^{th}_i$ and $F_i$ is minimized. Tonegawa et al. (2022) demonstrated using synthetic waveforms that the accurate *Δg* can be obtained

by this approach by numerical simulations using the locations of the DONET1 stations.

## 4 Results

### 4.1 Clock deviation

Because stations of SHM1, SHM6c, and SHM7c are relatively close to DONET1 or DONET2, direct and scattered waves can

be retrieved in the CCFs. Consequently, as shown in Fig. 3, sufficient measurements of clock deviations can be obtained for

these OBSs. The clock deviations obtained at stations SHM1, SHM3, SHM4, SHM6c, and SHM7c are approximated by linear

interpolation before and after the observations, since the median values are close to 0 s over the entire observation period. On

the other hand, the median values of SHM2 and SHM5 are slightly away from 0 s during the observation period, which

indicates non-linear clock deviations (Figs. 3 and S1). The average of the first and third quartiles of the clock deviations for

all the stations are ±0.03 s (Table 1).

**Table 1: Clock deviations averaged over October 2019–September 2020**

| Station code | First quartile (s) | Median (s) | Third quartile (s) |
| --- | --- | --- | --- |
| SHM1 | −0.0206 | 0.0046 | 0.0261 |
| SHM2 | −0.0259 | −0.0010 | 0.0236 |
| SHM3 | −0.0260 | −0.0035 | 0.0203 |
| SHM4 | −0.0257 | 0.0012 | 0.0276 |
| SHM5 | −0.0207 | 0.0052 | 0.0320 |
| SHM6c | −0.0165 | 0.0023 | 0.0197 |
| SHM7c | −0.0237 | −0.0048 | 0.0175 |

### 4.2 Spatio-temporal changes in *dv/v* and CC

Figures 4a and 4b show the temporal variations in *dv/v* at node KMC and OBSs, respectively. The *dv/v* changes at node KMC

shows an abrupt reduction of 0.1 % during the slow earthquake activity, while such reductions cannot be observed in the station

pairs of the OBSs. This fact indicates that the *dv/v* changes do not expand to the area of the OBSs. The reductions in *dv/v* at

node KMC continued, with a gradual recovery to the original *dv/v* values until the end of the observation period. Figures 4c



and 4d display the temporal variations in CC at node KMC and OBSs, respectively, and CC reductions are detected during the slow earthquake activity in both the areas. In particular, the troughs of the CC variations in the OBS pairs are slightly shifted, which indicates that they occur first in the eastern part of the survey area (SHM6c_SHM7c and SHM6c_SHM5) and later in the central part. These reductions were almost recovered to the original levels of CC immediately after the slow earthquake activity. Another CC reduction can be observed in the middle of 2020 for 3 pairs of two OBSs at node KMC.

The spatio-temporal variations in *dv/v* for Periods (1–6) are displayed in Fig 5. Substantial variations in *dv/v* are not observed in Period (1), whereas *dv/v* reduction can be observed in the eastern part of the survey area during Periods (2–6). This result is consistent with the results obtained at node KMC (Fig. 4), where *dv/v* reduction is not recovered to the original level after the event. The spatiotemporal variations in *Δg* for Periods (1–6) are displayed in Fig 6. Large localized *Δg* reductions are imaged at the eastern part of the survey area in Periods (2–5), and relatively weak *Δg* reductions occurred at the western part in the

same period. In addition, the *Δg*-reduced region at the eastern part in Period (5) migrated to the central part in Period (6). Figure 7 shows the spatiotemporal relationship between *Δg* and tremor distributions (Tamaribuchi et al., 2022); here, tremors with estimated errors ≤ 0.1º from the original catalog are preserved, and hence 84% of the tremors are excluded. Energy rates that radiated from tremors occurred at the middle of December were high, and these tremors occurred near the location of the maximum *Δg* reduction. The CC reductions in the eastern and western parts are merged on January 15–20, 2021, and substantial

reductions cannot be observed in February 2021.



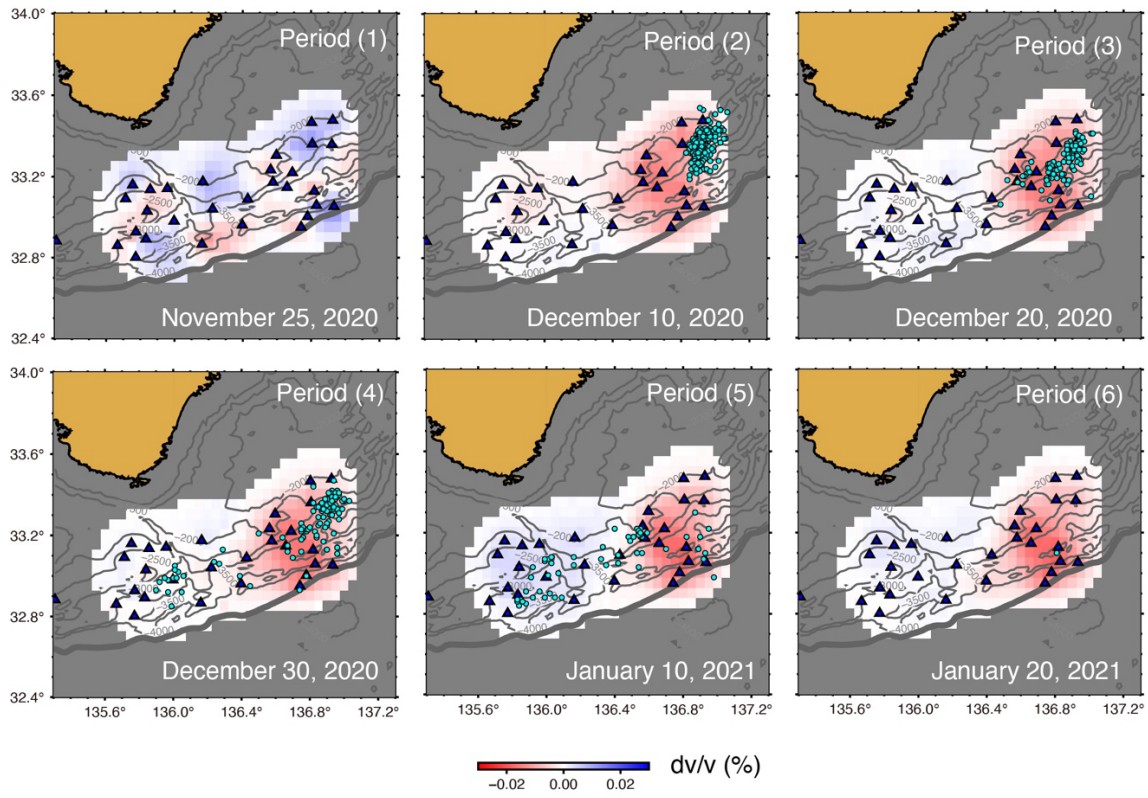

**Figure 5: Spatio-temporal *dv/v* variations in Periods (1–6). Circles represent the tremor locations within ±5 days from the reference dates, as determined by Tamaribuchi et al. (2022), from which tremors with spatial errors ≤0.1° are plotted (16% of tremors from the original catalog). Triangles shows the locations of the used stations.**






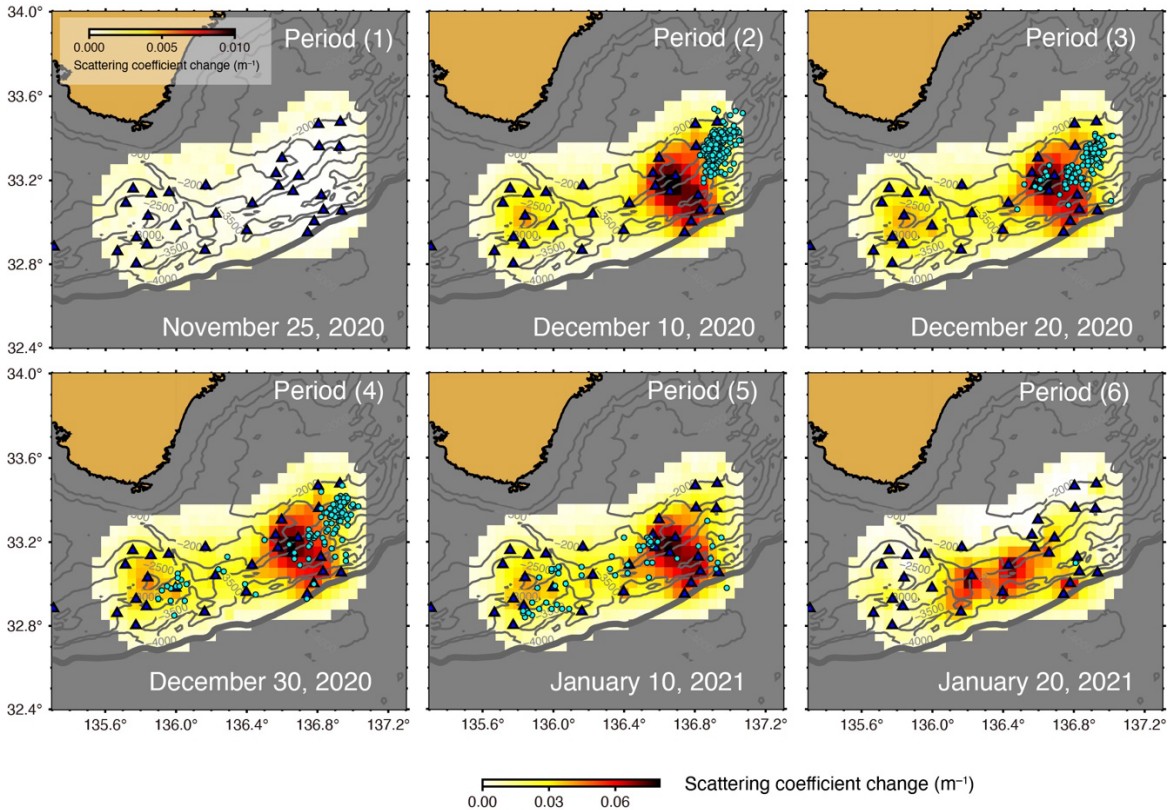

**Figure 6: Spatio-temporal scattering coefficient change (*Δg*) variations in Periods (1–6). Circles represent the tremor locations within ±5 days from the reference dates, determined by Tamaribuchi et al. (2022), from which tremors with spatial errors ≤0.1° are plotted (16% of tremors from the original catalog). Triangles shows the locations of the used stations. The amplitude in Period (1) is low.**

## 5 Discussion

### 5.1 Spatio-temporal relationship of *dv/v* and SCC(*Δg*) with tremors

A localized area of the *dv/v* reduction is observed in the eastern part of the survey area (Fig. 5). The *dv/v* reduction and tremor activities started from Period (2). Although the location of the maximum *dv/v* reduction is only at a small distance from the high activity area of tremors during Periods (2–3), the *dv/v* reduction area cover the tremor distribution. The tremor activity area migrated westward in Periods (4–5) and reached the central and western parts of the survey area; however, there is no considerable change in the *dv/v* reduction area. These facts indicate that the areas of *dv/v* reduction and tremors are correlated only at the early stages of the episode.

The *Δg* reduction area is also localized in the eastern part of the survey area, and the extent of localization is higher than that of the *dv/v* reduction area. The *Δg* reduction was initiated at the timing of the tremor activity in Period (2). Although the



location of the maximum *Δg* reduction slightly shifted westward from the tremor activity area, the *Δg*-reduced area surrounding the maximum *Δg* reduction cover the area of the tremors as well as that of the *dv/v* reduction. The tremor area moved to the location of the maximum *Δg* reduction in Periods (3–4). The areas of tremors moved westward in Period (5), but the CC reduction area did not change. However, in Period (6), the CC reduction area moved westward approximately 10 days after the westward tremor migrations observed in Period (5) (Figs. 6 and 7).

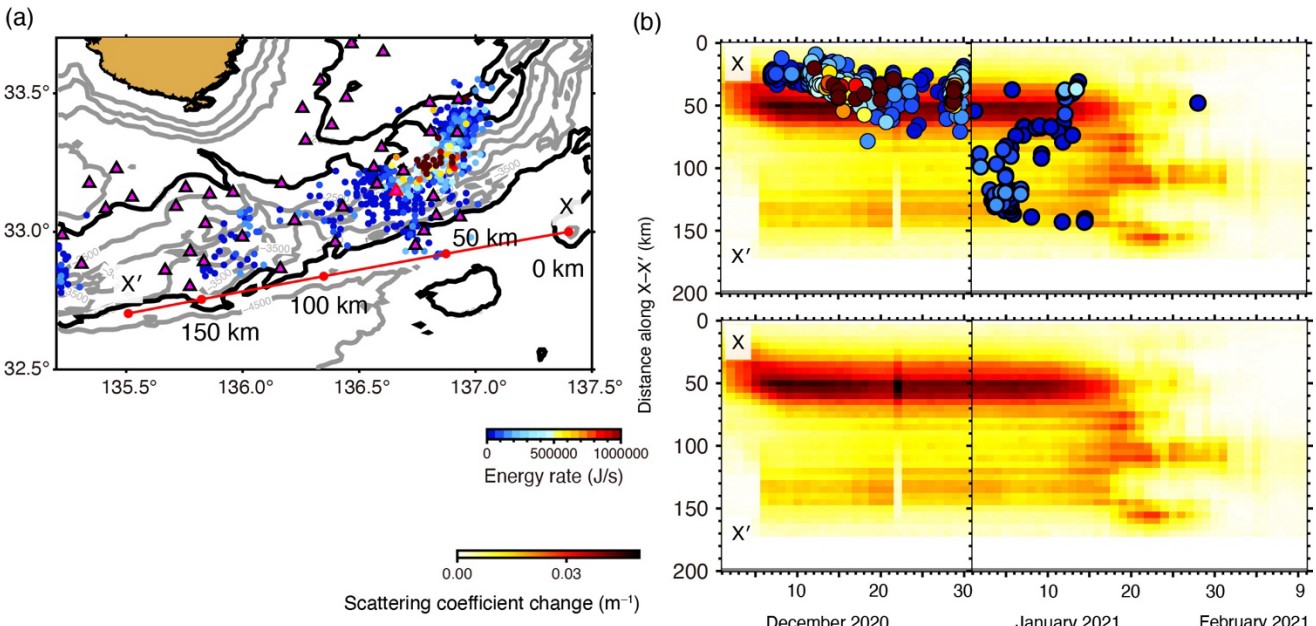

**Figure 7: Spatiotemporal scattering coefficient change (*Δg*) variations with tremors. (a) Tremor distribution (Tamaribuchi et al., 2022) and the line X–X′ at which the *Δg* and tremors are projected. (b) Tremors and *Δg* projected onto the line X–X′ in panel (a).**

## 5.2 Interpretation of temporal change difference in *dv/v* and SCC(*Δg*)

Our results indicate that the *dv/v* reduction shows a step-like variation at the time of the slow earthquakes, whereas the CC changes are recovered to the original level after the slow earthquake activity. Because CC reductions were observed during SSEs without or less tremor activities, the effects of the tremor signal contaminations on CC measurements appear minor (Tonegawa et al., 2022). In addition, although a step-like *dv/v* reduction was observed in this area due to the strong seafloor motions of the 2016 off-Mie earthquake (Mw6.0) that occurred beneath the DONET1 (Ikeda and Tsuji, 2018; Tonegawa et al., 2022), such strong motions were not observed during this episode. Furthermore, although slow earthquake activity with tremors and VLFEs occurred in October 2015 (e.g., Annoura et al. 2017; Baba et al., 2020), *dv/v* reductions were not observed during this event (Ikeda and Tsuji, 2018; Tonegawa et al., 2022). Therefore, it appears that the *dv/v* and CC changes obtained in this study were a result of different factors that occurred within the accretionary prism.



A candidate for the observed *dv/v* reduction is the change in crack conditions due to the deformation within the accretionary prism. Prior to the slow earthquake activity, the compressional stress in the trough-normal direction is dominant in the

accretionary prism, and cracks with a long axis oriented in the trough-normal direction are prevalent (e.g., Crampin 1977; 1981). Slow earthquakes at the shallow plate interface generate extensional stress within the overlying plate, which results in the extensional deformation of the accretionary prism along the trough-normal direction. Such deformations induce changes in the aspect ratio of the pre-existing cracks normal to the trough axis, and new cracks parallel to the axis are created. The updated conditions of the stress field and cracks are preserved with a small degree of recovery after the slow earthquakes. The

obtained *dv/v* changes are spatially limited to the eastern part of the survey region owing to the concentrated occurrence of tremors here (Tamaribuchi et al. 2022; Ogiso and Tamaribuchi, 2022), and the deformation of the sediment also appears to be concentrated in this region. Thus, this mechanism supports the step-like reduction for *dv/v*.

In contrast, the CC reductions are recovered to the original level after the slow earthquake activity. Tonegawa et al. (2022) interpreted that the CC reduction and its recovery are caused by transient upward fluid migrations at local scales. Before slow

earthquakes, the pore fluid pressure at the source region tends to increase with the fluid supply from the underlying oceanic crust through dehydration reactions (e.g., Kameda et al., 2011) and the accretionary prism itself by the tectonic compression (Saffer and Bekins, 1998). Although the uncertainties in the tremor depths are still high, the horizontal spatial coincidence between the VLFEs and the low velocity layer within the accretionary prism has been observed (Tsuji et al. 2014; Tonegawa et al., 2022; Akuhara et al., 2020). Several studies have also investigated whether tremor occurrences are related to the low

velocity layer (Hendriyana and Tsuji, 2021; Fahrudin et al., 2022; Akuhara et al., 2023b). The low velocity layer has a thickness of ~1 km and a horizontal distance of ~15 km in the dip direction (Park et al., 2010). Because a high pore-pressure region with a width of several hundred meters was found by a seafloor drilling program (Hirose et al., 2021), it seems that the low velocity layer consists of many such high pore-pressure volumes. Thus, it appears that fluids are trapped by impermeable caprocks in localized areas with low seismic velocities. Numerical simulations with hydraulic models also indicate the importance of

impermeable zones to locally increase pore pressures (Kaneki and Noda, 2023). A previous study has reported that temporal changes in seismic velocity and anisotropic structures occurred during the rupture of a slow-slip patch in the Hikurangi subduction zone (Zal et al., 2020). When slow earthquakes occurred at the end of 2020, the caprocks fractured, and the trapped fluids migrated upwards. Our results of CC reductions reflect such fluid migration passing through shallow depths. In particular, the CC reductions in Period (6) are observed approximately 10 days after the tremor activities in the central part (Figs. 7 and

8c), which may correspond to the fracture-induced fluid migrations.

To increase pore fluid pressures at the source region, mechanisms that trap fluids within the accretionary prism are required, and the topographic relief of the top surface of the oceanic crust may be invoked for the fluid trapping. The topographic relief induces deformation at the downdip side of the relief (Fig. 8b). Such deformations intensively form horizontally sheeted clay minerals in and around the deformation area immediately above the oceanic crust, which can create impermeable caprocks.

As a result, fluids cannot escape laterally or vertically and are trapped below the impermeable sheets; eventually the pore pressure becomes higher than that in the surrounding regions. Such fluid-trapped areas would have been created locally in





several parts of the Nankai subduction zone, which may be related to the sporadic occurrences of the VLFEs. Indeed, VLFEs occur on the downdip sides of the topographic relief (Shiraishi et al. 2020; Hashimoto et al. 2022). Moreover, several discontinuous reflections with high impedance contrasts are imaged within the low velocity zone (Tsuji et al., 2014), which

may represent impermeable sheets due to relief-induced deformations.

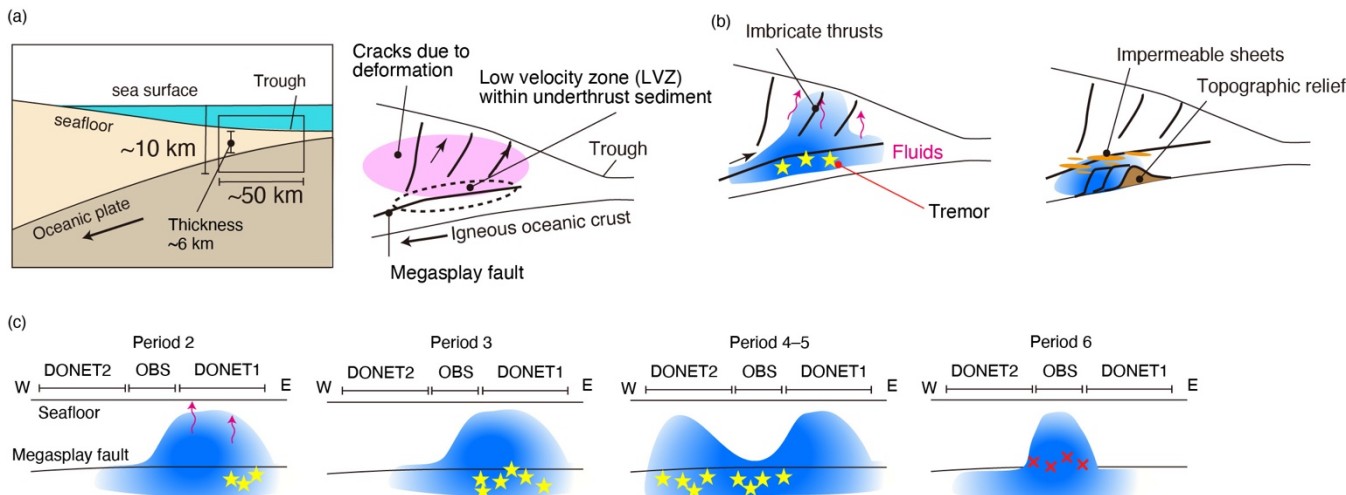

**Figure 8: Interpretation for our results with tremor and fluid distributions. (a) Illustration for crack creations due to the deformation causing step-like reduction in *dv/v*. (b) (left) Upward fluid migration during slow earthquakes in the cross section along the dipping**
**direction. (right) Fluids are trapped by the impermeable sheets that are created by the deformation by topographic reliefs. (c) Spatio-temporal distributions of tremors and upward fluid migrations in the cross sections along the strike direction. Crosses indicate the locations of caprock fractures.**

## 6 Conclusion

We present the temporal variations in *dv/v* and CC associated with the slow earthquake activity that started at the end of 2020 in the Nankai subduction zone. The *dv/v* variations exhibit the step-like reductions at several pairs of stations that are located in the eastern part of the survey area. These features are considered to be constructed by changes in the aspect ratios of pre-existing cracks and/or newly created cracks due to the sediment deformation induced by the slow earthquake episode. The CC variations show the transient reductions at the eastern and western parts of the survey area, which are spatially correlated with

the tremor distribution. It is considered that the CC reductions can be linked to the transient fluid migration from the source region of the slow earthquakes. Thus, *dv/v* and CC are sensitive to different factors in the subseafloor structure, and we suggest that both the *dv/v* and CC variations be monitored for understanding the slow earthquake generations and their influence on the surrounding structure.




**Data availability**

DONET data can be downloaded from the website operated by National Research Institute for Earth Science and Disaster Resilience (NIED) (https://doi.org/10.17598/NIED.0008). *dv/v* and CC data used in this study can be downloaded from a repository Zenodo (https://zenodo.org/records/14020538).

**Author contribution**

T.T processed the data, and drafted the manuscript. T.A, S.T, T.Ts edited the manuscript and contributed to the interpretation. T.A, Y.Y, H.S, M.S aimed to acquire the data. All authors contributed to the final version of the manuscript.

**Competing interests**

The authors declare that they have no conflict of interest.

**Acknowledgements**

This work was supported by JSPS KAKENHI Grant No. JP21H05202 in Scientific Research on Transformative Research Areas "Science of Slow-to-Fast earthquakes" and Cooperative Research Program of Atmosphere and Ocean Research Institute, The University of Tokyo (Research Vessel Shinsei-maru, SH19-27 and S21-31.

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
