# Peer review of "Accretionary prism deformation and fluid migration caused by slow earthquakes in the Nankai subduction zone"

_EGUsphere, 2024_

## Referee Comment (RC1)

Dear editors and authors:

Tonegawa et al. studied temporal changes in seismic velocity and waveform correlation (scattering property) in the Nankai Trough subduction zone. They interpreted that changes in velocity and scattering property of the sediment occurred through different mechanisms: the first was sediment deformation and the second was fluid migration. Data analysis was carried out appropriately and conclusions are generally valid. The current manuscript would benefit from implementing the suggestions that I describe in detail below. Most of these are merely about presentation, but there are other comments where an additional explanation might be needed to better understand the authors' findings.

- Line 248 "This fact indicates that the dv/v changes do not expand to the area of the OBSs.": In Fig. 5, are not SHM6c and SHM7c included in the velocity reduction area? In Fig. 4, no clear velocity reduction is observed for this pair.
- 2. According to Fig. 6, the maximum value of  $\Delta g$  estimated in this study is approximately 0.08 m-1. Therefore, given that  $g_0=1/\ell=1/10.8$  km,  $\Delta g/g_0\sim 0.0864\%$ . This value is significantly smaller than the range estimated in previous studies (several % to 100%) (Obermann et al., 2013; Obermann et al., 2014; Hirose et al., 2023).

Obermann, A., Planès, T., Larose, E., & Campillo, M. (2013). Imaging preeruptive and coeruptive structural and mechanical changes of a volcano with ambient seismic noise. Journal of Geophysical Research, [Solid Earth], 118(12), 6285–6294. https://doi.org/10.1002/2013jb010399

Obermann, A., Froment, B., Campillo, M., Larose, E., Planès, T., Valette, B., et al. (2014). Seismic noise correlations to image structural and mechanical changes associated with the M w 7.9 2008 Wenchuan earthquake. Journal of Geophysical Research, [Solid Earth], 119(4), 3155–3168. https://doi.org/10.1002/2013jb010932

Hirose, T., Wang, Q. Y., Campillo, M., & Nakahara, H. (2023). Time-lapse imaging of
seismic scattering property and velocity in the northeastern Japan. Earth and Planetary
ScienceScienceLetters.Retrievedfromhttps://www.sciencedirect.com/science/article/pii/S0012821X23003345

- 3. Sections 4.2 and 5.1 " $\Delta g$  reduction" and " $\Delta g$ -reduced": Since  $\Delta g$  represents the change in the scattering coefficient, these expressions might be misleading. Consider changing " $\Delta g$ -reduced region" to "large  $\Delta g$  region" and "maximum  $\Delta g$  reduction" to "maximum  $\Delta g$ ".
- 4. Lines 212 and 222: Consider adding references for Equations 5, 6, and 7.
- 5.  $\Delta g/t$  in Equation 6 ->  $\Delta S(1/t)(dv/v)_actual K$  (e.g., Obermann et al., 2014).
- 6. Please add  $\Delta S$  to the right-hand side of Equation 7.

---

## Referee Comment (RC2)

**Accretionary prism deformation and fluid migration caused by slow earthquakes in the Nankai subduction zone**

**Summary:**

Tonegawa et al. discusses the connection between tremors (slow earthquakes) and the fluid migration in the Nanaki subduction zone (Japan). The focus lies on changes in the seismic velocity and correlation coefficient at the subduction zone by combining a temporally installed OBS network and a permanently installed cabled OBS network. The study period contains a series of slow earthquakes starting in the northwestern part and moving southeast ward. The correlation coefficients (cc) show a similar movement whereas the dv/v remains stable in the northwest. Both measurements (dv/v and cc) are used to monitor the subsurface (structure and properties). It is interpreted that the step-like drop in dv/v and the temporally drop of the cc point toward aspect rotation of cracks and fluid migration.

**General Comments:**

Generally, the present preprint is close to the published paper Tonegawa et al. 2022. To me, it has to be clear which are the differences between the two publications (data, method, and findings). It has to be immediately clear what we lean by including the temporally OBS network. Further, section 3.4 "Spatial mapping of velocity change and seismic scattering coefficient" is very similar to Tonegawa et al. 2022. Instead of repeating all this, just focus on the new parts and refer to Tonegawa et al. 2022 for the rest.

Thorugout the manuscript, the terms "tremor, slow eq, slow slip event, VLFE" seem sometimes to be used interchangeable. Pleas introduce all of them carefully by explicitly mentioning the difference between them and further, used them carefully. As a non-expert on this topic, I found it difficult to follow and I believe the manuscript would benefit from a clearer distinction between these terms.

The understandability mainly in the Introduction and Discussion can be improved by slightly restructuring the sections.
- E.g. l32-44: You start introducing the Nankai subduction zone and the different types of seismic (and aseismic) events). Then you move toward the different types of observations and at the end, it feels like you finish the introduction of the Nankai subduction zone from the event perspective.

- E.g. l55-63: You talk about the Nankai subduction zone twice, once about the central part and the second time about the western part. In between you talk about the Hikurangi subduction zone.
- E.g. l131-134: A sentence about the CCFs between OBS and DONET followed by a sentence about DONET clocks and then again, a sentence about CCFs between OBS and DONET. L150-152 is again about the station pairs. This should be said in l131 and following.
- E.g. l194: First sentence about stations and following sentence is talking about tremor activity.
- E.g. l314-317: Sentence explaining mechanism followed by sentence about special occurrence followed by sentence about observation of mechanism in data. The middle sentence separates a logical thought.

Be consistent in using the term OBS. Sometimes it is used for the temporally deployed OBS network but sometimes it is also used for cabled sensors. It is clear to me that both are OBS but to improve readability I would not talk from OBS in connection with the cabled array and instead just use the station name or node name.

If I understand correctly, there are data from the temporally installed OBS from September 2019 until May 2021. I would appreciate to see the whole (or at least longer) dv/v and CC time-series (Fig. 4, 5, 6).

The terms north, south, east and west, all are used but I think you are mainly talking about two regions, the one around KMD and MRD. If this is correct, I would choose eighter east-west or maybe north-south but avoid using both.

**Line Comments:**

**Introduction:**

L 15: "heterogeneous structures (correlation coefficient, CC)", more is not needed in the abstract

L17: spatial or temporal pattern? Please specify

L26: "(1) the seismic velocity"

L42: "Nankai Trough, they are approximately 1–5 years" to "Nankai Trough between 1 to 5 years"

L44: add citation after first sentence

L65-66: I would avoid the term OBS in connection with DONET since you call the temporal network OBS. Maybe just write "Using the continuous record of the Dense Oceanfloor Network system for Earthquakes and Tsunamis (DONET, Fig. 1) …"

L68: Change the comma to a point "… subduction zone. Tonegawa et …"

L71: "However, the time after the slow…"

L73: I don't know the term "seismic structure", therefore I would replace it with "subsurface structure"

L74-76: "In this study, we include the temporary OBS deployments into the existing DONET deployment to cover the gap between DONET1 and DONET2 and estimate temporal changes in the subsurface structure using ambient noise correlation techniques. Additionally, we investigate the connection between fluids and the occurrence of slow earthquakes."

L76-78: This sentence is not needed. I hope that you choose a period with slow eq. to study them. Please make sure that after deleting this sentence, you connect the sentence before and after (e.g. do not take first about slow eq. and then about tremor)

L78-80: "The tremor activity started on December 5, 2020, peaked on December 11 and 28 and faded by early February 2021 (Fig. 1b)."

L80-84: Not needed, everything already mentioned

**Data:**

L96: "We used the continuous vertical component records from 49 DONET stations."

L96-98: In Figure 1 it looks more like the tremors occur in the northern part of the network.

L102-104: "To fill the gap, 15 OBSs with short-period sensors (1Hz) were deployed between September 2019 and May 2021. 10 OBSs were used for two subarrays. In this study, we used 2 OBSs located at the center of the two subarrays. In total 7 OBSs were used."

**Methods:**

L106-110: This is not needed

L112-121: "We calculated cross-correlation functions (CCFs) from ambient noise records following established methods (e.g. Campillo and Paul, 2003; Shapiro et al., 2005; Brenguier et al., 2007). To suppress energetic signals, such as those from earthquakes, we applied lognormal-shaped functions (Tonegawa et al., 2020; 2022). A bandpass filter of

0.5–2.0 Hz combined with spectral whitening was used. In the chosen frequency band, the ambient seismic noise is dominated by acoustic-coupled Rayleigh (ACR) waves. These waves propagate through the ocean and the entire accretionary prism in this region with 1.3–1.5 km/s (Tonegawa et al., 2015). CCFs were computed using 600-second time windows, and a 30-day moving average was obtained by stacking daily CCFs for each station pair. If the duration affected by the lognormal-shaped function was less than 70% of the 30-day period, the corresponding CCF was discarded. Additionally, reference CCFs for each station pair were generated by stacking CCFs over the entire observation period. In the CCFs, direct ACR wave propagation between station pairs appears at early lag times, while scattered waves are observed in the coda (Fig. 2)."

L141: It seems like you have an example for this symmetrical shift. Add such an example figure to the Supplement and add a reference. If figure 2.b shows this asymmetry, add a clear statement to the figure caption and add a reference to figure 2.b.

L142: "– 20 – 20 s" looks like there are two minus signs. I would replace the second one with the symbol used in "2-s" so that it is clearer. (Also e.g. l163)

L143-145: I cannot follow. This sentence starts with CFs and ends with CCFs.

L145-147: "If, within a given time window, the cross-correlation coefficients of the CCFs exceed 0.9 for more than 85% of the observation period, the window is considered to contain coherent signals throughout the period. The delay times of these CCFs are then interpreted as clock deviations."

L163: $dv/v = -dt/t$ should be introduced in the sentence before. Possible reference: Poupinet et al., 1984 (Monitoring velocity variations in the crust using earthquake doublets: An application to the Calaveras Fault, California)

L163-165: I do not understand this sentence.

L174: What do you think is the reason that MRE-OBS is below 70%?

L193-194: Why southern part? The figures 5 and 6 show variations in the northeast.

L195-200: I would like to see more periods (earlier and later). Further, I think it is a bit problematic to use 30 d data and show periods with an increment of just 10 days. To me this is too much overlap. If 30 d data used to get stable results, reduce the number of periods between November 25 2020, and January 20 2021.

L200-230: To me these formulas do not have to be in the main text since they have been published already in Tonegawa et al., 2022. Move them to the supplement. The last paragraph with information about the parameters can still be part of the main text if preferred.

**Results:**

L235-240: Very clear and interesting. To me, the key message is that some station clocks can be interpolated linearly and some not but table 1 does not support this finding. Therefore, I suggest moving the table to the supplement. Further, the clock of station SHM3 could also be interpreted as slightly periodic instead of linear.

L246-248: Also include the DONET2 nodes in these sentences to support the last sentence.

L251-254: I cannot see the time offset in CC time series for OBSs (Fig 4.d). You are analyzing the spatio-temporal scattering changes later (Fig 6) so you don't have to say anything about the CC values and their spatio-temporal evolution here.

L254: This is confusing because here you use the term OBS for the cable array.

L567: Is it also consistent with other nodes shown in supplement figure?

L262-264: Correct but the weaker (less energy) tremors are also located at the same spot. The energy rates are not used to support your findings.

L264: I'm not sure if the CC reductions "merged" or if the strong one form the eastern part moved and the weaker one in the western part disappeared. I think your time resolution is not high enough to distinguish the two options.

**Discussion:**

L284-285: By expanding the study period it would become clearer how strong the dv/v reduction and the tremor occurrence really correlate and if there is something like healing after the tremor series.

L291: You stacked over 30 d, can you make a statement about 10 d?

L299: I would call it a step-like drop.

L302: Drop "In addition"

L304: Which episode do you mean? Drop "Furthermore"

L308-311: Are these the cracks shown in Fig. 8a? If so, add a reference.

L311-314: Is this shown in one of the subplots in Fig 8? Might be helpful.

L314: Does this mean you expect a slight increase of dv/v after the slow eq but not back to the original level? Do you see this?

L317: I do not understand what is meant by "this mechanism" because you explain the spatial distribution in the sentence before and not a mechanism. I think what you mean is that the step-like drop with almost no healing supports the crack thesis. If so, I the sentence beforehand discussing the spatial occurrence of the dv/v drop should not split the mechanism and the observed drop (sentence before and after).

L318-333: This is introduction and too much background for a discussion. The amount of background information should be shortened and more closely connected to your new findings. Further, make clear what the new findings about tremor location and fluid migration are compared to Tonegawa et al 2022.

L336-345: Concept is clear and supported by references. But to me it is not clear how this new analysis supports this concept. (I think this does not have to be added here and you can leave this last paragraph how it is but make this clear in the two antecedent paragraphs.)

L308-345: I understand that dv/v is linked with cracks and CC with fluids. For the increased pore pressure, you need this impermeable clay sheets (to trap fluids). How are these two things (cracks and high pore pressure) linked? Intuitively I would think that more cracks (dv/v) drop lead to a pressure decrease and not increase. I find this very interesting but I cannot completely follow here.

**Conclusion**

L355-356: "We present the temporal variations in dv/v and CC in the Nankai subduction zone associated with the slow earthquake activity that began at the end of 2020."

**Figure Comments:**

Figure 1:
- add a and b and replace top and bottom
- (a) legend for seismic network
- Label the Nankai Trough with 0 km
- Remove station names and numbers and add figure like Tonegawa et al. 2022 Fig 1 to Supplement
- Add (thicker) black edge color for stations used in this study
- (inserted map of Japan) color ocean blue or land brown and change plate boundaries to orange (like in (a))
- (b) I would remove the vertical black lines marking the end of the month
- Caption:
  - DONET1 and DONET2 are circles and squares vs temp. OBS are triangles
  - Time format: Please double check the time format and keep it consistent (December 1$^{st}$ 2020 is not the same as December 1, 2020)
  - "The histogram represents the number of tremors per day whereas the red line shows the cumulative number of tremors between December 1, 2020 and February 5, 2021. Tremor locations are represented in (a)."

Figure 2:
- Caption:
  - Time format
  - "CCFs for XX station pairs stacked between START-DATE and END-DATE (1 year). The two lines represent a propagation velocity of 1.5 km/s."
  - (b) add the time frame for stacking

Figure 3:
- (a) Nanaki Trough (0 km) + 10 km orange (like Fig. 1 a)
- (a) Add legend explaining red, yellow and pink markers therefore remove "The symbols are the same as those in Fig. 1."
- (b) Cut off the uppermost 100 days (600-700 seems to be empty)
- (b) Make black dots transparent that density of points becomes visible
- (b) "The estimated clock deviations at SHM7c represented by the black dots. The red line marks the median value and right respectively left cyan line indicate the first respectively third quartiles."

Figure 4:
- OBS is not the same as temporally deployed OBS
- The y-axis do not make sense since you plot the station pair with offset.

Figure 5:
- Color bar can be larger

- Use the same shapes for the different arrays as in Figure 1 and 2 (circles, triangles, squares)
- Would be interesting to expand this analysis in time (backward and forward)
    o Maybe the subsurface needs just more time to "relax" after the slow eq. activity
- I know that sometimes stacking over long time periods is necessary to acquire stable dv/v results. But here, to me there is too much time overlap between the different periods. If there is no way to shorten the stacking period, the number of presented periods should be reduced to avoid that much overlap.
- You write in L174 that you could not use the node MRE but MRE stations are plotted on the map. Just plot the stations used to generate the dv/v anomaly or make the unused stations transparent.

Figure 6:
- No extra color bar is needed for the first Period
- See comments Figure 5

Figure 7:
- Not sure why energy rates are plotted. I don't see a correlation between energy rates and scattering coeff.
- Color bar can be larger
- Use the same shapes for the different arrays as in Figure 1 and 2 (circles, triangles, squares) → if tremors are black, use same colors for stations as in Figure 1 and 2
- Unclear why figure (b) appears twice. Upper plot is sufficient.

Figure 8:
- What is the difference between e.g. (c) Period 2 and 3 when there are (no) pink arrows pointing upward?
- Slow eq vs. tremor
- Just use a,b,c,.. and remove the left vs right

---

## Author Response (AR1)

Reviewer #1

Dear editors and authors:

Tonegawa et al. studied temporal changes in seismic velocity and waveform correlation (scattering property) in the Nankai Trough subduction zone. They interpreted that changes in velocity and scattering property of the sediment occurred through different mechanisms: the first was sediment deformation and the second was fluid migration. Data analysis was carried out appropriately and conclusions are generally valid. The current manuscript would benefit from implementing the suggestions that I describe in detail below. Most of these are merely about presentation, but there are other comments where an additional explanation might be needed to better understand the authors' findings.

[Response] We appreciate careful reading and the constructive comments from the reviewer #1. In particular, owing to the comment, we were able to correct the equations.

1. Line 248 "This fact indicates that the dv/v changes do not expand to the area of the OBSs.": In Fig. 5, are not SHM6c and SHM7c included in the velocity reduction area? In Fig. 4, no clear velocity reduction is observed for this pair.

[Response] The sensitivity kernel area for the pair of SHM6c_SHM7c covers near the margin of the velocity reduction area (Fig. R1), so it seems that the result from the pair do now show velocity reductions.

[Figure]

Fig. R1 The left panel is the same as the panel for Period (3) in Fig. 5. The right panel is the sensitivity kernel for the station pair of SHM6c and SHM7c.

2. According to Fig. 6, the maximum value of $\Delta g$ estimated in this study is approximately 0.08 m-1. Therefore, given that $g0=1/l=1/10.8$ km, $\Delta g/g0\sim0.0864\%$. This value is significantly

smaller than the range estimated in previous studies (several % to 100%) (Obermann et al., 2013; Obermann et al., 2014; Hirose et al., 2023).

[Response] T.To is sorry for confusing the reviewer #1 for this point, and mistakes the unit. The correct unit is km**(–1) not m**(–1). The unit in Figs. 6 and 7 is corrected.

3. Sections 4.2 and 5.1 "$\Delta g$ reduction" and "$\Delta g$-reduced": Since $\Delta g$ represents the change in the scattering coefficient, these expressions might be misleading. Consider changing "$\Delta g$-reduced region" to "large $\Delta g$ region" and "maximum $\Delta g$ reduction" to "maximum $\Delta g$".

[Response] Thank you very much raising for this point. We changed the description accordingly.

4. Lines 212 and 222: Consider adding references for Equations 5, 6, and 7.

5. $\Delta g/t$ in Equation 6 -> $\Delta S(1/t)(dv/v)$_actual K (e.g., Obermann et al., 2014).

6. Please add $\Delta S$ to the right-hand side of Equation 7.

[Response 4-6] We appreciate these comments. We cited references and modified these equations. Also, according to the comment from the reviewer #2, we moved this part to Supplement.

Reviewer #2

Summary:

Tonegawa et al. discusses the connection between tremors (slow earthquakes) and the fluid migration in the Nanaki subduction zone (Japan). The focus lies on changes in the seismic velocity and correlation coefficient at the subduction zone by combining a temporally installed OBS network and a permanently installed cabled OBS network. The study period contains a series of slow earthquakes starting in the northwestern part and moving southeast ward. The correlation coefficients (cc) show a similar movement whereas the dv/v remains stable in the northwest. Both measurements (dv/v and cc) are used to monitor the subsurface (structure and properties). It is interpreted that the step-like drop in dv/v and the temporally drop of the cc point toward aspect rotation of cracks and fluid migration.

General Comments:

Generally, the present preprint is close to the published paper Tonegawa et al. 2022. To me, it has to be clear which are the differences between the two publications (data, method, and findings). It has to be immediately clear what we lean by including the temporally OBS network. Further, section 3.4 "Spatial mapping of velocity change and seismic scattering coefficient" is very similar to Tonegawa et al. 2022. Instead of repeating all this, just focus on the new parts and refer to Tonegawa et al. 2022 for the rest.

Throughout the manuscript, the terms "tremor, slow eq, slow slip event, VLFE" seem sometimes to be used interchangeable. Pleas introduce all of them carefully by explicitly mentioning the difference between them and further, used them carefully. As a non-expert on this topic, I found it difficult to follow and I believe the manuscript would benefit from a clearer distinction between these terms.

The understandability mainly in the Introduction and Discussion can be improved by slightly restructuring the sections.

[Response] We appreciate careful checks and valuable comments for the contents of this manuscript. Owing to the comments from the reviewer #2, the manuscript was greatly improved. In particular, the estimations of clock deviations are further constrained by the comment.

To distinguish this manuscript from Tonegawa et al. (2022), as the reviewer #2 suggested, we inserted a sentence (L76), and moved the method for mapping dv/v and delta g to Supplement (Section 3.4).

For terms of slow earthquakes, we defined them in Introduction (L33).

E.g. l32-44: You start introducing the Nankai subduction zone and the different types of

seismic (and aseismic) events). Then you move toward the different types of observations and at the end, it feels like you finish the introduction of the Nankai subduction zone from the event perspective.

[Response] We modified Introduction section accordingly, with reflecting the comments from the reviewer #2. We explain the Nankai subduction zone and different types of slow earthquakes (L31–45), introduce the relationship between fluid and slow earthquakes (L46–75), and document the event that is targeted in this study (L76–82).

E.g. l55-63: You talk about the Nankai subduction zone twice, once about the central part and the second time about the western part. In between you talk about the Hikurangi subduction zone.

[Response] In L31–46, we introduce the slow earthquakes in the Nankai subduction zone, and their relations to fluids observed in various subduction zones are also introduced in L47–56. In L57–65, we state that some evidences of fluid migrations observed in various subduction zones. This should be stated here because the main target of this study is fluid migration and deformation of the accretionary prism, and we would like to keep the current sentence structure.

E.g. l131-134: A sentence about the CCFs between OBS and DONET followed by a sentence about DONET clocks and then again, a sentence about CCFs between OBS and DONET. L150-152 is again about the station pairs. This should be said in l131 and following.

[Response] Thanks. We slightly modified the sentences L130 and L148.

E.g. l194: First sentence about stations and following sentence is talking about tremor activity.

[Response] Yes. Because we define the timeframes in the following sentence, it is important to state here our targeted period when tremor activity occurred. We slightly changed the description (L206).

E.g. l314-317: Sentence explaining mechanism followed by sentence about special occurrence followed by sentence about observation of mechanism in data. The middle sentence separates a logical thought.

[Response] Thanks. We slightly edited this part (L311).

Be consistent in using the term OBS. Sometimes it is used for the temporally deployed OBS network but sometimes it is also used for cabled sensors. It is clear to me that both are OBS but to improve readability I would not talk from OBS in connection with the cabled array and

instead just use the station name or node name.
If I understand correctly, there are data from the temporally installed OBS from September 2019 until May 2021. I would appreciate to see the whole (or at least longer) dv/v and CC time-series (Fig. 4, 5, 6).

[Response] In the revised manuscript, we use the station and node names to DONET stations, and OBS to temporary OBSs. Thanks for this comment, Also, we extended the analyzed dates for dv/v and delta g mapping in Figs. S5 and S6. Also, please see our replies for the subsequent comments.

The terms north, south, east and west, all are used but I think you are mainly talking about two regions, the one around KMD and MRD. If this is correct, I would choose eighter east-west or maybe north-south but avoid using both.

[Response] We removed the terms of north and south, except for "southwestern Japan" in the first sentence of Introduction, in the revised manuscript.

Line Comments:
Introduction:
L 15: "heterogeneous structures (correlation coefficient, CC)", more is not needed in the abstract

[Response] We changed it (L15).

L17: spatial or temporal pattern? Please specify

[Response] Here, temporal is correct, so we changed it (L17).

L26: "(1) the seismic velocity"

[Response] Thanks. We changed it (L26).

L42: "Nankai Trough, they are approximately 1–5 years" to "Nankai Trough between 1 to 5 years"

[Response] We changed it (L45).

L44: add citation after first sentence

[Response] The relation between fluids and slow earthquake generations is described in the following sentences. Here, we would like to just put this sentence without citations, and cite relevant references at the following sentences.

L65-66: I would avoid the term OBS in connection with DONET since you call the temporal network OBS. Maybe just write "Using the continuous record of the Dense Oceanfloor Network system for Earthquakes and Tsunamis (DONET, Fig. 1) …"

[Response] Thanks for this suggestion. We changed it accordingly (L67).

L68: Change the comma to a point "… subduction zone. Tonegawa et …"

[Response] Thanks (L69).

L71: "However, the time after the slow…"

[Response] We changed it accordingly (L73).

L73: I don't know the term "seismic structure", therefore I would replace it with "subsurface structure"

[Response] We changed it accordingly (L77).

L74-76: "In this study, we include the temporary OBS deployments into the existing DONET deployment to cover the gap between DONET1 and DONET2 and estimate temporal changes in the subsurface structure using ambient noise correlation techniques. Additionally, we investigate the connection between fluids and the occurrence of slow earthquakes."

[Response] Thanks for nice rewording. We changed to the description (L76–78).

L76-78: This sentence is not needed. I hope that you choose a period with slow eq. to study them. Please make sure that after deleting this sentence, you connect the sentence before and after (e.g. do not take first about slow eq. and then about tremor)

L78-80: "The tremor activity started on December 5, 2020, peaked on December 11 and 28 and faded by early February 2021 (Fig. 1b)."

L80-84: Not needed, everything already mentioned

[Response] We modified this part accordingly (L76–82).

Data:

L96: "We used the continuous vertical component records from 49 DONET stations."

[Response] We changed it. (L94)

L96-98: In Figure 1 it looks more like the tremors occur in the northern part of the network.

[Response] To avoid confusing, we changed from "from the southern part" to "near the

Nankai Trough". (L95)

L102-104: "To fill the gap, 15 OBSs with short-period sensors (1Hz) were deployed between September 2019 and May 2021. 10 OBSs were used for two subarrays. In this study, we used 2 OBSs located at the center of the two subarrays. In total 7 OBSs were used."
Methods:
[Response] We modified this sentence (L100).

L106-110: This is not needed
[Response] We deleted this part (L103–104).

L112-121: "We calculated cross-correlation functions (CCFs) from ambient noise records following established methods (e.g. Campillo and Paul, 2003; Shapiro et al., 2005; Brenguier et al., 2007). To suppress energetic signals, such as those from earthquakes, we applied lognormal-shaped functions (Tonegawa et al., 2020; 2022). A bandpass filter of 0.5–2.0 Hz combined with spectral whitening was used. In the chosen frequency band, the ambient seismic noise is dominated by acoustic-coupled Rayleigh (ACR) waves. These waves propagate through the ocean and the entire accretionary prism in this region with 1.3–1.5 km/s (Tonegawa et al., 2015). CCFs were computed using 600-second time windows, and a 30-day moving average was obtained by stacking daily CCFs for each station pair. If the duration affected by the lognormal-shaped function was less than 70% of the 30-day period, the corresponding CCF was discarded. Additionally, reference CCFs for each station pair were generated by stacking CCFs over the entire observation period. In the CCFs, direct ACR wave propagation between station pairs appears at early lag times, while scattered waves are observed in the coda (Fig. 2)."
[Response] The readability was greatly improved by this suggestion (L105–117).

L141: It seems like you have an example for this symmetrical shift. Add such an example figure to the Supplement and add a reference. If figure 2.b shows this asymmetry, add a clear statement to the figure caption and add a reference to figure 2.b.
[Response] A good example would be Fig. 1 in Brenguier et al. (2008), so we cited the paper at this part (L137). A similar figure is also displayed in Fig. 2b in Tonegawa et al. (2022). Brenguier et al. (2008), Nature Geoscience, https://doi.org/10.1038/ngeo104

L142: "– 20 – 20 s" looks like there are two minus signs. I would replace the second one with the symbol used in "2-s" so that it is clearer. (Also e.g. l163)

[Response] Minus, hyphen, and dash are defined by the character code, so the description is correct. However, to be honest, T.T agrees with this comment, and we change to "and" here (L139).

L143-145: I cannot follow. This sentence starts with CFs and ends with CCFs.
[Response] (L140 in the revised manuscript) In this technique, to measure dv/v and CC, we calculate cross-correlation functions between two CCFs (reference and individual) from ambient noises, and we would like to define CF for cross-correlating two CCFs.

L145-147: "If, within a given time window, the cross-correlation coefficients of the CCFs exceed 0.9 for more than 85% of the observation period, the window is considered to contain coherent signals throughout the period. The delay times of these CCFs are then interpreted as clock deviations."
[Response] Thanks a lot. We changed the sentence accordingly (L142).

L163: dv/v = - dt/t should be introduced in the sentence before. Possible reference: Poupinet et al., 1984 (Monitoring velocity variations in the crust using earthquake doublets: An application to the Calaveras Fault, California)
[Response] We moved dv/v=–dt/t to the previous sentence (L164), and cited Poupinet et al. (1984) (L165).

L163-165: I do not understand this sentence.
[Response] We modified this sentence (L167).

L174: What do you think is the reason that MRE-OBS is below 70%?
[Response] In the previous manuscript, we removed station pairs including MRE21. T.T is sorry for confusing the reviewer #2. We looked at 1-day CCFs for a pair of MRE21–MRE18 (Fig. R2), and found that 1-day CCFs for a couple of days during the slow earthquake activity have large amplitudes and these CCFs reduce the number of available 30-day CCFs and 20-day CCFs. Although we checked the raw waveforms with a bandpass filter of 0.5–2.0 Hz, we did not find any specific large signals related to tremors. Perhaps, ACR waves propagating at long distance may be contaminated in the ambient noises, and this influences the resulting CCFs. However, why such effects are particularly found at MRE21 is still unclear, and is the subject to future studies.

[Figure]

Fig. R2 1-day (left) and 30-day (right) CCFs for the station pair of MRE21 and MRE18. 445–485 days correspond to the slow earthquake activity (December 5, 2020–January 14, 2021).

L193-194: Why southern part? The figures 5 and 6 show variations in the northeast.

[Response] We changed the description (L206), and added this point in the captions of Figs. 5 and 6.

L195-200: I would like to see more periods (earlier and later). Further, I think it is a bit problematic to use 30 d data and show periods with an increment of just 10 days. To me this is too much overlap. If 30 d data used to get stable results, reduce the number of periods between November 25 2020, and January 20 2021.

[Response] For temporal variation results (Figs. 5, 6, and 7), we use the 20-day CCFs and the reliability of CC from the 20-day CCFs is confirmed by Fig. S4.

L200-230: To me these formulas do not have to be in the main text since they have been published already in Tonegawa et al., 2022. Move them to the supplement. The last paragraph with information about the parameters can still be part of the main text if preferred.

[Response] We moved this part to Supplement.

Results:

L235-240: Very clear and interesting. To me, the key message is that some station clocks can be interpolated linearly and some not but table 1 does not support this finding. Therefore, I suggest moving the table to the supplement. Further, the clock of station SHM3 could also

be interpreted as slightly periodic instead of linear.

[Response] We moved Table 1 to Supplement as Table S1. Owing to the comment from the reviewer #2, Figs. 3 and S2 are modified, and the results from SHM1, SHM2, SHM3, SHM5 and SHM7c show substantial clock deviations. We added this explanation (L219).

L246-248: Also include the DONET2 nodes in these sentences to support the last sentence.

[Response] We added a sentence for DONET2 (L232).

L251-254: I cannot see the time offset in CC time series for OBSs (Fig 4.d). You are analyzing the spatio-temporal scattering changes later (Fig 6) so you don't have to say anything about the CC values and their spatio-temporal evolution here.

[Response] We added Fig. S4c (L238), which zoomed up the slow earthquake period of Fig. 4d. The time offset can be observed in even 30-day CCFs. It is important to state in Result section that such time offsets can be observed in the original data, and this helps reader to understand the discussion of the spatio-temporal variations.

L254: This is confusing because here you use the term OBS for the cable array.

[Response] We modified this sentence accordingly (L239).

L567: Is it also consistent with other nodes shown in supplement figure?

[Response] Maybe yes. We added a possibility for the station pairs in the text (L239).

L262-264: Correct but the weaker (less energy) tremors are also located at the same spot. The energy rates are not used to support your findings.

[Response] According to the comment from the Reviewer #2, we removed the information of energy rates of tremors from Fig. 7.

L264: I'm not sure if the CC reductions "merged" or if the strong one form the eastern part moved and the weaker one in the western part disappeared. I think your time resolution is not high enough to distinguish the two options.

[Response] We modified Figs. 5–7 with 20-day CCFs. In Fig. 7b, we added arrows that indicate large delta g at the eastern and western parts are combined.

Discussion:

L284-285: By expanding the study period it would become clearer how strong the dv/v reduction and the tremor occurrence really correlate and if there is something like healing

after the tremor series.

[Response] We added Figs. S5 and S6 for dv/v and delta g at non-slow-earthquake periods, and added a paragraph that explains their spatio-temporal variations.

L291: You stacked over 30 d, can you make a statement about 10 d?

[Response] We compare dv/v in Fig. S4a for and CC in Fig. S4b for 10-, 20-, and 30-day CCFs. CC from 20-day and 30-day CCFs is consistent but it from 10-day CCFs shows that more stacking periods are needed. dv/v for all the stacking periods are consistent. We added this explanation (L178–183).

L299: I would call it a step-like drop.

[Response] We change the description (L294).

L302: Drop "In addition"

[Response] We deleted it (L297).

L304: Which episode do you mean? Drop "Furthermore"

[Response] It is the slow earthquake episode that began from the end of 2020. We changed the descriptions (L293 and 298).

L308-311: Are these the cracks shown in Fig. 8a? If so, add a reference.

[Response] We refer to Fig. 8a here (L304). Thanks.

L311-314: Is this shown in one of the subplots in Fig 8? Might be helpful.

[Response] We added Fig. 8b.

L314: Does this mean you expect a slight increase of dv/v after the slow eq but not back to the original level? Do you see this?

[Response] Yes. Thanks. We added this explanation (L310).

L317: I do not understand what is meant by "this mechanism" because you explain the spatial distribution in the sentence before and not a mechanism. I think what you mean is that the step-like drop with almost no healing supports the crack thesis. If so, I the sentence beforehand discussing the spatial occurrence of the dv/v drop should not split the mechanism and the observed drop (sentence before and after).

[Response] Because we added one sentence (L310), we modified the sentence including

"mechanism" (L314).

L318-333: This is introduction and too much background for a discussion. The amount of background information should be shortened and more closely connected to your new findings. Further, make clear what the new findings about tremor location and fluid migration are compared to Tonegawa et al 2022.

L336-345: Concept is clear and supported by references. But to me it is not clear how this new analysis supports this concept. (I think this does not have to be added here and you can leave this last paragraph how it is but make this clear in the two antecedent paragraphs.)

[Response for the two comments] Thanks for raising this point. The CC results in this study are consistent with those from Tonegawa et al. (2022), although the studied area is wider. In L316–346, we would like to suggest a possible scenario for upward fluid migration that intermittently occurs, and the scenario is originated from the observations in this study. To explain the observations, the descriptions of fluid storage, upward migrations, and trapping are needed. It seems that such arguments can be included in the Discussion section. Also, in L318, we define the positioning of this part.

L308-345: I understand that dv/v is linked with cracks and CC with fluids. For the increased pore pressure, you need this impermeable clay sheets (to trap fluids). How are these two things (cracks and high pore pressure) linked? Intuitively I would think that more cracks (dv/v) drop lead to a pressure decrease and not increase. I find this very interesting but I cannot completely follow here.

[Response] This is a good question. It seems that, although the background permeability distribution caused by geological conditions, including clay sheets, is significantly altered by regular and slow earthquakes through fractures, it is slightly fluctuated by crack conditions that are influenced by the stress field. This is just a qualitative explanation, and should be confirmed by other approaches, e.g., numerical simulations, which is beyond the scope of this study.

Conclusion

L355-356: "We present the temporal variations in dv/v and CC in the Nankai subduction zone associated with the slow earthquake activity that began at the end of 2020."

[Response] We changed it accordingly (L357).

Figure Comments:
Figure 1:

add a and b and replace top and bottom

(a) legend for seismic network

Label the Nankai Trough with 0 km

Remove station names and numbers and add figure like Tonegawa et al. 2022 Fig 1 to Supplement

Add (thicker) black edge color for stations used in this study

(inserted map of Japan) color ocean blue or land brown and change plate boundaries to orange (like in (a))

(b) I would remove the vertical black lines marking the end of the month

Caption: DONET1 and DONET2 are circles and squares vs temp. OBS are triangles

Time format: Please double check the time format and keep it consistent (December 1st 2020 is not the same as December 1, 2020)

"The histogram represents the number of tremors per day whereas the red line shows the cumulative number of tremors between December 1, 2020 and February 5, 2021. Tremor locations are represented in (a)."

[Response] We modified Fig. 1. Because the Nankai Trough is not 0 km, we corrected the caption. Also, we added Fig. S1 that show the station names.

Figure 2:

Caption: Time format

"CCFs for XX station pairs stacked between START-DATE and END-DATE (1 year). The two lines represent a propagation velocity of 1.5 km/s."

(b) add the time frame for stacking

[Response] We changed the caption accordingly.

Figure 3:

(a) Nanaki Trough (0 km) + 10 km orange (like Fig. 1 a)

(a) Add legend explaining red, yellow and pink markers therefore remove "The symbols are the same as those in Fig. 1."

(b) Cut off the uppermost 100 days (600-700 seems to be empty)

(b) Make black dots transparent that density of points becomes visible

(b) "The estimated clock deviations at SHM7c represented by the black dots. The red line marks the median value and right respectively left cyan line indicate the first respectively third quartiles."

[Response] Owing to the comment from the reviewer #2, we changed the method to estimate the clock deviations, and modified this figure and the caption.

Figure 4:

OBS is not the same as temporally deployed OBS

The y-axis do not make sense since you plot the station pair with offset.

[Response] We modified the figure.

Figure 5:

Color bar can be larger

Use the same shapes for the different arrays as in Figure 1 and 2 (circles, triangles, squares)

Would be interesting to expand this analysis in time (backward and forward) Maybe the subsurface needs just more time to "relax" after the slow eq. activity

I know that sometimes stacking over long time periods is necessary to acquire stable dv/v results. But here, to me there is too much time overlap between the different periods. If there is no way to shorten the stacking period, the number of presented periods should be reduced to avoid that much overlap.

You write in L174 that you could not use the node MRE but MRE stations are plotted on the map. Just plot the stations used to generate the dv/v anomaly or make the unused stations transparent.

[Response] We modified this figure. Also, we added more panels in Figs. S5.

Figure 6:

No extra color bar is needed for the first Period

See comments Figure 5

[Response] We modified this figure. Also, we added more panels in Fig. S6. The color bar used in Period (1) is different from that for Periods (2–6). This is also noted in the caption.

Figure 7:

Not sure why energy rates are plotted. I don't see a correlation between energy rates and scattering coeff.

Color bar can be larger

Use the same shapes for the different arrays as in Figure 1 and 2 (circles, triangles, squares)

→ if tremors are black, use same colors for stations as in Figure 1 and 2

Unclear why figure (b) appears twice. Upper plot is sufficient.

[Response] We modified this figure accordingly.

Figure 8:

What is the difference between e.g. (c) Period 2 and 3 when there are (no) pink arrows pointing upward?

Slow eq vs. tremor

Just use a,b,c,.. and remove the left vs right.

[Response] We modified this figure.

---

## Author Response (AR2)

Dear the editorial support team, Copernicus Publications in Solid Earth,

Thank you very much for your time and effort to our manuscript.

In our manuscript, we do not use initials in the title page, so we do not change them. However, in the final manuscript and Supplement, we only changed the affiliations of Yusuke Yamashita, because he moved to another university on April 1st, 2025.

Sincerely,

Takashi TONEGAWA

Japan Agency for Marine-Earth Science and Technology

3173-25, Showa-machi, Kanazawa-ku, Yokohama, 236-0001, Japan

TEL: +81-45-778-5965

E-mail: tonegawa@jamstec.go.jp